# Quantifying the Third-Party Loss in Building Construction Sites Utilizing Claims Payouts: A Case Study in South Korea

**Ji-Myong Kim [1]**, **Kag-Cheon Ha [2]**, **Sungjin Ahn [1]**, **Seunghyun Son [3]** and **Kiyoung Son [4],***

[1] Department of Architectural Engineering, Mokpo National University, Mokpo 58554, Korea; jimy6180@gmail.com (J.-M.K.); sunahn@mokpo.ac.kr (S.A.)
[2] Department of Police & Fire Administration, U1 University, Yeongdong 29131, Korea; hkc723@u1.ac.kr
[3] Department of Architectural Engineering, Kyung Hee University, Suwon 17104, Korea; seunghyun@khu.ac.kr
[4] School of Architectural Engineering, University of Ulsan, Ulsan 44610, Korea
[*] Correspondence: sky9852111@ulsan.ac.kr

**Abstract:** This study aims to quantify the losses to third-parties on construction sites by determining the loss indicators and identifying the relationship between the losses and the indicators to improve the sustainability on building construction sites. The growing size and intricacy of recent construction projects have resulted in the growth of losses, both in quantity and frequency. Notably, third-party losses are rapidly increasing owing to the urbanization of the environment and increases in construction scale. Therefore, for efficient and sustainable construction management, a financial loss assessment model is essential to mitigate and manage such loss. This study uses the third-party losses on construction sites obtained from a major South Korean insurance company to describe the difference from the material losses and to disclose the loss indicators based on actual economic losses. ANOVA analysis and multiple regression analysis are adopted to identify the variance and define the loss indicators and to make prediction models, respectively. Several groups of loss indicators are investigated, including construction information and the occurrence of natural disasters. The findings and results of this research afford an essential guide to sustainable construction management, and they can serve as a first stage loss assessment model for construction projects.

**Keywords:** sustainable construction management; third-party loss; loss assessment; loss indicators; ANOVA; multiple regression analysis

## 1. Introduction

To avoid reliance on only expert insight and experience, the present risk management paradigm of construction projects requires quantified models. The risk of construction projects has grown in both quantity and frequency of occurrence, in tandem with the rising scale and complexity of current construction projects. For example, the Korea Occupational Safety and Health Agency (KOSHA) reported that, in the 20-year period of 1995–2014, the average frequency rate of injury on construction sites was 3.50. The Insurance Statistics Information Services (INSIS) have stated that the total sum of losses in construction increased annually. The total losses in 2014 were about 49.2 billion KRW, which is 5.1 times larger than the total losses in 1995 of 9.6 billion KRW.

Moreover, rapid urbanization in many Asian countries has not only led to an increase in construction projects but also to a surge in the risk of construction projects. In general, construction projects in urban and metropolitan regions are more risky and complicated than construction projects in rural regions, owing to marginal circumstance [1]. The reason for this is that urban and metropolitan

regions are encompassed by established utility lines, buildings, and traffic facilities. Hence, to increase sustainable construction management, present construction management and loss prediction models of construction site need more sophisticated and scientific methodology, and they also need to take into account the third-party losses that are direct or indirect losses from the impact of construction activities.

### 1.1. Loss Assessment of Third-Party for Sustainable Construction Management

A third-party in a construction project is defined as an outsider or organization not directly related to the construction project, such as contractors, subcontractors, and workers [2]. Therefore, third-party damage denotes damage to the body or property of a third-party caused by a construction project; e.g., physical injury, damage to tangible and intangible assets, and destruction [3]. For example, third-party might include commercial, industrial, and agricultural workers around the site and their property and pedestrians around the site. Most of the previous studies on risk management and safety at construction sites have centered on workers or structures at the construction sites, and analysis and research of third-party damage are infrequent [4–7]. However, damage to a third-party can lead to a number of problems, such as suspension of construction, administrative penalties, and financial compensation, as well as secondary losses, such as a decrease in work productivity due to disputes and adverse effects on the company's reputation. This is a necessity of management because the results of efforts to increase profitability, such as shortening the construction period and saving budget through construction management, are lost due to the unexpected damage. Therefore, many construction project managers have improved and developed advanced risk assessment and management methods to reduce this potential damage, but they have focused on accidents occurring within the boundaries of construction sites [8]. Therefore, for more comprehensive and sustainable construction site risk management, a management system that includes damage to third-parties occurring outside of the construction site is required.

A wide range of companies are developing models to assess the loss of construction sites to handle the remarkable and significant losses that occur during construction. The companies recognize that it is crucial to identify and measure risk according to scientific examination for the construction management of sustainable construction sites. Additionally, through this model, the risks of a construction site can ultimately be managed and reduced.

Insurance companies and insurance brokers have generated in-house loss assessment models to estimate potential loss. For example, the major reinsurance companies and insurance brokers, Munich Reinsurance Company, Swiss Reinsurance Company, and Willis Re, have created their standard models such as the Munich Re-Engineering Expert Tool (MRET) and the Project Underwriting Management Application (PUMA). These loss assessment models help insurance company engineering underwriters and clients to understand and calculate the latent risks in construction. On the other hand, the use of assessment models is narrowed to insurance issues because the models are intended for use in the insurance business considering prevailing market customs [9]. Additionally, because these models are only allowed to be used by their employees or by a limited number of customers, there is the disadvantage that they are not readily available to the public. Therefore, it is difficult for the general public to evaluate the third-party loss on a construction site.

Furthermore, many vendors (e.g., Risk Management Solution, Applied Insurance Research, and EQECAT$^{TM}$) are providing models to assess construction risks [10,11]. Even if the models are designed for worldwide use, it is problematic to automatically use these models for building construction projects in some countries.

This is because the variations in regional vulnerability in building construction, and in the scale and frequency of natural hazards, can lead to an increase in the uncertainty of the prediction results, and errors can result from models that are not specific to the particular country involved. This makes a difference to the third-party loss, which is heavily influenced by the surrounding environment. Third-party losses are losses that are incurred outside of the construction site, so differences arise in different countries due to the various natural settings and building codes. Additionally, the

users of these models cannot adjust the models to reflect their risk capital, risk appetite, and other vital characteristics of projects because the indicators and calculation logic of the models are in an impenetrable black box. Therefore, users of the model hardly understand how material losses and third-party loss are generated due to their hidden logic. Consequently, to evaluate elaborate third-party losses, there should be research that can be easily accessed by the general public and that reflects national third-party risks.

Researchers have conducted numerous studies of loss assessment methods and losses in construction. Nevertheless, although these studies have identified loss indicators, there are still shortages of synthetic and quantitative studies for third-party loss. Kim (2009) determined the indicators by each phase of a construction project based on both interviews of field managers and on previous studies [12]. He categorized the critical factors as business situations, natural hazards, geographic conditions, and environmental issues. Kim (2008) developed a method to estimate construction risks and defined indicators such as contractor workability, site location, project information, and natural hazard experiences [13]. Based on these factors, he created a risk grading checklist and weighted factors using literature reviews and expert interviews. Park (2005) conducted expert interviews and several case studies to define the latent risk in construction projects [14]. Lee et al. (2003) proposed a methodology to quantify the amount of risk utilizing the construction company's workability and specific construction information [15].

These studies have found that loss assessment models should include a diversity of indicators. The reason for this is that the risk is not exclusively determined because it is a compound of vulnerability, hazard, and exposure [16]. However, there is still a lack of quantitative studies. The findings of these studies are from quantitative research, literature reviews, and expert surveys. Furthermore, these studies do not distinguish between material loss and third-party loss, so it is challenging to assess third-party loss alone. For a scientific and objective analysis of third-party losses, it is essential to quantify the magnitude of the risk through statistical analysis of third-party losses. Analyzing and identifying the feature of losses using a quantitative method is the initial step of third-party loss assessment model development. Quantitative loss examination that includes various categories of loss indicators utilizing statistical analysis is critical to developing an efficient third-party loss assessment model. Therefore, this study compares third-party loss and material loss through ANOVA analysis, and it identifies significant indicators through multiple regression analysis.

### 1.2. Research Goal and Objectives

The goal of this study is to assess the importance of third-party loss and loss indicators in construction sites through the multiple regression modeling of historical loss data. The objectives are to investigate the third-party and material loss records that arise in real building construction sites and to identify the relationship between third-party losses and associated loss indicators utilizing a multiple regression in South Korea as a case study. Research hypotheses are as follows: (1) There will be a difference between the loss of the material and the loss of a third-party that occurs at the construction site; (2) Third-party loss can be predicted through various indicators.

Construction site losses from an insurance company are collected to reveal the actual economic losses. Third-party losses collected and analyzed in this study mention dividends for losses paid to third-parties in insurance record data. The findings and results of this study can be adopted and used as vital guidelines for the initial phase of developing financial loss assessment models for construction projects in South Korea. Moreover, these guidelines can be applied to other countries that, in their construction environment and natural disasters, bear some similarities to South Korea.

### 1.3. Research Methodology

The subject of study is the loss amount of insurance companies for the collection of quantified data and statistical analysis occurring at the construction site. The collected data was classified into two types of loss to exam the difference between third-party loss and material loss. The analysis tool of

this study used the IBM Statistical Package for the Social Sciences (SPSS) V23 to compare the two loss groups and analyze their relationships with the indicators.

The detailed research procedure is as follows. This study uses six steps to compare the third-party loss and material loss, and to define the relationships between the loss and the indicators. First, previous research was investigated to obtain academic reviews of the relevant indicators and assessment models. Second, the losses of an insurance company occurring at various construction sites were collected as the dependent variable. Third, we divided the losses into two categories, separated into third-party or material. For example, we categorized any losses that occurred in the field as material loss and considered any third-party losses that happened outside the field from construction activities as third-party loss. Fourth, the losses in the two categories were compared utilizing ANOVA. Fifth, several groups of indicators were collected, for example, the construction information and the occurrence of natural disasters, as a comprehensive study. Last, a multiple regression analysis approach was employed to statistically examine the association of the dependent variables and loss indictors.

## 2. Data Collection

This study adopts a loss record of Contractors' all-risks insurance (CAR). CAR is designed to cover the entire scope of unanticipated losses during all stages of a construction project for contractors. The coverage is for building construction, as well as for new and supplemental building construction. CAR covers losses incurred by contractors related to losses, including any losses from a third-party, substantive losses to machines and tools, and demolition costs. Some losses are ruled out if the losses are included in the CAR disclaimer [17]. Analysis of claim payouts is one of the best ways to represent financial losses in construction, owing to the high-resolution of the data and the externalization of costs [18]. Quantified insurance data is especially useful because the losses are independently investigated for each case with detailed information, including loss detail and project information, and losses are decided through an impartial process, following examinations conducted by both engineers and claims adjusters.

The losses are divided into two groups: third-party loss and material loss. For example, we classify a loss occurring in a construction objective or a construction site as material loss and a loss outside of a construction site as third-party loss.

*Data Availability*

This study gathered the losses of a major insurance company of CAR in South Korea for the period 2001–2016. The total number of cases was 430. Of these, there were 120 material losses and 310 third-party losses. The data used in this study was very limited in use. Due to the nature of the insurer's data, even if no customer information was included, there was no public access in order to prevent possible problems. Owing to the nature of the data, the amount of loss is the net amount of losses paid by the insurance company, which does not take into account any insurance conditions. No kind of personal information about the insured was included. The claim payout records received from the insurance company involved information such as the date of the accident, location, occupancy, structure type, construction period, floor, underground, loss amount, and description of loss. The scope of this study is limited to the Republic of Korea.

## 3. Data Analysis

*3.1. Descriptive Statistics and ANOVA Analysis*

This study adopts the IBM Statistical Package for the Social Sciences (SPSS) V23 for the analysis. Table 1 shows that the frequency of third-party losses was about 2.6 times higher than the material losses. Furthermore, comparing the average, the severity of third-party losses was about 4.5 times higher than that of the material losses. Therefore, the third-party losses were more critical than the material losses in both frequency and severity.

**Table 1.** Descriptive statistics for losses.

| Category | Number | Percentage of the Total | Min. | Max. | Mean | Std. Deviation |
|---|---|---|---|---|---|---|
| Material | 120 | 28.0 | 0.13 | 236.67 | 15.01 | 29.98 |
| Third-party | 310 | 72.0 | 0.18 | 1686.86 | 67.52 | 161.61 |

Nevertheless, statistical validation was required to analyze the significant difference between the losses, i.e., third-party and material, of construction sites as a systematic approach.

This study adopted ANOVA analysis to analyze the differences between the two groups. If the two groups show significant differences between their average loss ratios, it will prove possible to take advantage of the logical risk management per material and third-party when performing construction management of future construction sites.

The ratio was calculated as the cost of the losses (KRW) divided by the total cost of construction (KRW). The null hypothesis was that there was no significant difference between the average loss ratios of third-party and material:

$$H0 : \mu_{ml} = \mu_{tl} \tag{1}$$

$$H1 : \mu_{ml} \neq \mu_{tl} \tag{2}$$

where $\mu_{ml}$ is the loss ratio of material loss and $\mu_{tl}$ is the loss ratio of third-party loss.

Table 2 verifies that the null hypothesis was rejected because the *p*-value of 0.005 was smaller than 0.05. This concludes that the average of material and third-party losses incurred at the construction site are different.

**Table 2.** Result of ANOVA analysis.

| | Sum of Squares | Degree of Freedom | Mean Square | F | Sig. |
|---|---|---|---|---|---|
| Between Groups | 0.002 | 1 | 0.002 | 8.037 | 0.005 |
| Within Groups | 0.082 | 429 | 0.000 | | |
| Total | 0.084 | 430 | | | |

*3.2. Investigation of Third-Party Losses*

Based on the description of the third-party loss in the claim payout records, the losses were grouped into seven loss causes, such as failure of construction, typhoons, heavy rain, worker carelessness, flooding, vibration/noise/dust, and "etc." Table 3 represents the descriptive statistics for the causes of loss. Failure of construction, vibration/noise/dust, worker carelessness, and flooding were the most commonly occurring accidents, at frequencies of 74.8, 15.0, 5.0, and 2.0%, respectively. Typhoons, worker carelessness, heavy rain, and flooding were the most critical accidents, with average costs of 355.7, 150.1, 144.1, and 91.9 billion KRW, respectively.

Figure 1 represents the severity and frequency matrix by cause of loss. The horizontal axis represents the severity through the average insurance loss amount for each loss cause. The vertical axis represents the frequency through the proportion of total occurrences by cause of loss. Through the matrix, the frequency and severity of each loss cause are indicated, and through this, the characteristics of the loss cause can be defined by division into four zones according to the frequency and depth. In addition, through this technique, it is possible to consider the main causes of management loss [19]. For example, Figure 1 can be divided into four zones as listed: Zone 1, the severity and frequency are low; Zone 2, the severity is moderate but the rate is high; Zone 3, the incidence is low but the severity is high; and Zone 4, the severity and frequency are both high. Each zone can help supervisors and workers respond to potential causes of loss. For example, the cause of loss in Zone 2, failure of construction, has low severity with very high frequency, denoting that this type of loss can occur at any time on the site. Thus, both the supervisor and the laborer should pay more attention to this kind of failure. Typhoons are the singular cause of loss in Zone 3, with low frequency but with very

high severity. As a result, construction sites should prepare appropriate countermeasures to prevent and mitigate such loss, such as installing water gates and water pumps on-site and making detention ponds near the site.

**Table 3.** Descriptive statistics and description for causes of loss.

| Cause of Loss | Description | Number | Frequency (%) | Average (Bn. KRW) | Standard Deviation (Bn. KRW) | Max. (Bn. KRW) | Min. (Bn. KRW) |
|---|---|---|---|---|---|---|---|
| Failure of construction | Losses due to poor construction at the site | 232 | 74.8 | 79.2 | 173.3 | 1493.2 | 1.0 |
| Vibration/noise/dust | Losses from vibration, noise, and dust generated at the site | 47 | 15.0 | 90.6 | 108.6 | 467.7 | 1.1 |
| Worker carelessness | Losses caused by carelessness of workers at the construction site | 16 | 5.0 | 150.1 | 149.7 | 599.7 | 3.2 |
| Flooding | Losses triggered by flooding | 6 | 2.0 | 91.9 | 135.3 | 361.0 | 2.9 |
| Etc. | Other losses | 5 | 1.7 | 38.9 | 37.8 | 96.1 | 4.7 |
| Heavy rain | Losses produced by heavy rain | 4 | 1.3 | 144.1 | 9.2 | 152.1 | 131.8 |
| Typhoons | Losses caused by typhoon | 1 | 0.3 | 355.7 | 0.0 | 355.7 | 355.7 |

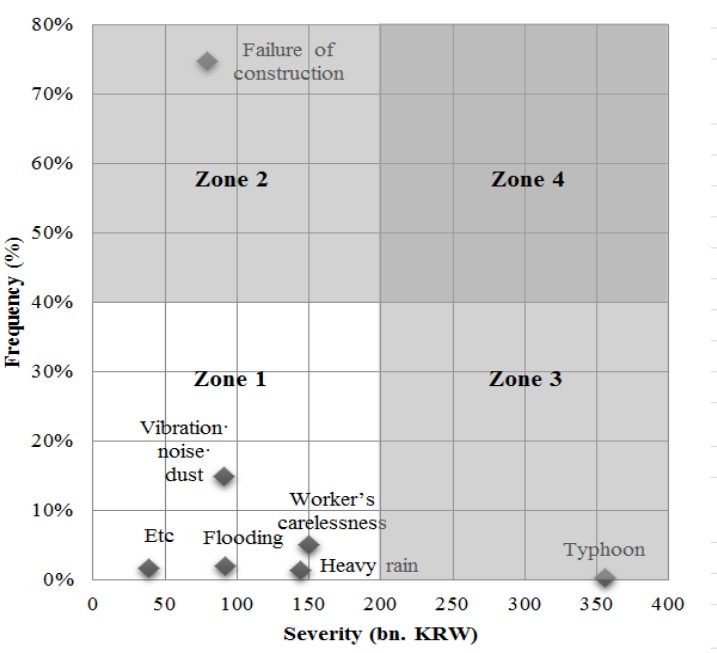

**Figure 1.** Severity and frequency matrix by cause of loss.

## 4. Multiple Regression Analysis

This study employs a multiple regression analysis approach to define the interrelationship among the losses and the indicators to create a loss estimation model according to the losses. This research only examines the losses of the third-parties to quantify the amount of risk and identify the significant loss indicators. The dependent variable, loss ratio (%), is the loss percentage. The loss ratio is the cost of third-party losses in building construction fields (KRW) divided by the cost of the construction project (KRW).

### 4.1. Indicators

This study divided the indicators into two categories (i.e., construction information and natural disaster) and selected indicators through a review of the previous construction project's loss quantification research and sustainable construction management [3,20–22]. Table 4 shows the indicators that were identified. The numerical construction information represents the level of intricacy, scale, and challenge of construction projects, and it can reflect the risk amount of construction [23]. For example, the scale of construction represents the loss amount as a loss indicator. Kim et al. (2015) found that financial losses and the scale of the building have a statistically significant relationship [24]. Ryu et al. (2016) proved that the number of floors and the numbers of underground floors are closely related to the losses incurred at the construction site [20].

**Table 4.** List of indicators for sustainable construction management.

| Category | Indicator | Explanation | Unit |
|---|---|---|---|
| Construction information | Structural type | Structural type of building (Dummy variable) | 1. Reinforced concrete 2. Other |
| | Total Month | Total length of construction period | Numeral (month) |
| | Floor | Number of floors | Numeral |
| | Underground | Number of underground floors | Numeral |
| Natural disaster | Tropical cyclone | Tropical cyclone risk at the site | Expected wind speed ((0–5), km/h) 0. Area 0: (74–141) 1. Area 1: (142–184) 2. Area 2: (185–212) 3. Area 3: (213–251) 4. Area 4: (252–299) 5. Area 5: ≥300 |

Moreover, we can describe the level of difficulty of construction by the type of structure. For example, the construction sites for reinforced concrete have more accidents and are more prone to disasters than are the construction sites for other sorts of construction [25].

This study characterizes the loss indicator of construction information by four indicators: structural type of building (reinforced concrete, other), total length of construction period, number of aboveground levels (including ground level), and number of underground levels.

Meteorological disasters can primarily affect the loss of construction [1,26]. For example, a construction site located in a hurricane-prone area often faces construction delays owing to the gusts and heavy rain caused by hurricanes. Therefore, the risk grade of natural disasters, e.g., tropical cyclone, flood, heavy snow, etc., undoubtedly represents the loss of construction [27–29].

This research adopts an indicator, the risk grade of tropical cyclones, to show the vulnerability from natural disasters. The study utilizes the Munich Reinsurance Company's natural hazard map, the Natural Hazards Assessment Network (NATHAN) World Map of Natural Hazards, to determine the risk grade. This online mapping system for global natural hazards was developed to accurately estimate the risk of natural disasters, such as earthquakes, floods, and tropical cyclones, at specific locations [30]. The risk grade of tropical cyclones is used to reflect the characteristics of natural disasters in South Korea. These risk grades are gathered based on the latitude and longitude coordinate information of each construction site.

### 4.2. Results

Table 5 shows the descriptive statistics of the independent variables and dependent variables. This study uses the backward elimination method to discover the best-fit regression model. Table 6 summarizes the regression model. The model is statistically significant because the *p*-value of 0.000 is

smaller than 0.05. The adjusted R-square value of 0.471 denotes that this relationship is clarified with a 47.1% margin of variance. The loss ratio, the dependent variable, is converted into a natural log.

**Table 5.** Descriptive statistics.

| Variable | Min. | Max. | Mean | Std. Deviation |
|---|---|---|---|---|
| Dependent | | | | |
| Loss ratio | 0.07 | 1686.86 | 66.72 | 160.67 |
| Independent | | | | |
| Structural type | 0.00 | 1.00 | 0.69 | 0.46 |
| Total months | 5.00 | 126.00 | 22.26 | 14.59 |
| Aboveground floors | 1.00 | 80.00 | 19.10 | 11.65 |
| Underground floors | 0.00 | 12.00 | 3.48 | 2.12 |
| Tropical cyclone | 1.00 | 4.00 | 1.40 | 0.86 |

**Table 6.** Coefficients of the model.

| Indicator | Coeff. | Std. Error | Beta Coeff. | $p > |z|$ | VIF |
|---|---|---|---|---|---|
| Constant | 3.827 | 0.308 | | 0.000 | |
| Structural type | 0.847 | 0.211 | 0.219 | 0.000 | 1.481 |
| Total months | −0.062 | 0.007 | −0.501 | 0.000 | 1.402 |
| Aboveground floors | −0.015 | 0.009 | −0.095 | 0.094 | 1.578 |
| Underground floors | −0.087 | 0.049 | −0.103 | 0.075 | 1.642 |
| Tropical cyclone | 0.168 | 0.099 | 0.08 | 0.091 | 1.119 |
| Number of observations | 310 | | | | |
| F | 65.901 | | | | |
| Adj-R2 | 0.471 | | | | |

Table 6 shows the coefficients of the regression model. There are five significant indicators of the loss ratio: (1) structural type, (2) total months, (3) aboveground floors, (4) underground floors, and (5) tropical cyclone. Each indicator is constructed to represent the loss ratio. The other indicators are excluded because the *p*-value is higher than 0.10. The variance inflation factors (VIF) values range from 1.119 to 1.642. The VIF values denote that the indicators have no critical multicollinearity.

The beta coefficients indicate the placing of the indicators regarding the volume of impact on the loss ratio scaled (0 to 1). Considering the weight of the coefficients, the placing is listed as follows: (1) total months, (2) structural type, (3) underground floors, (4) aboveground floors, and (5) tropical cyclone.

Based on these coefficients, as shown in Equation (3), a multiple regression model can be made through five indicators to predict the dependent variable.

$$\ln(\text{Loss ratio}) = 3.827 + (0.847 \times \text{Structural type}) + (−0.062 \times \text{Total months}) + (−0.015 \times \text{Aboveground floors}) + (−0.087 \times \text{Underground floors}) + (0.168 \times \text{Tropical cyclone}) + \varepsilon \tag{3}$$

### 4.3. Validity of the Model

The Kolmogorov–Smirnov value is utilized to test the normality of the residuals [31]. The values state that the residuals of the model are ordinarily distributed because the *p*-value of 0.081 is larger than 0.05, as seen in Table 7.

**Table 7.** Test of model normality.

|  | Kolmogorov–Smirnov | |
|---|---|---|
|  | **Statistic** | **Sig.** |
| Ln (Loss ratio) | 0.049 | 0.081 |

Moreover, the histogram of standardized residuals and the Q–Q plot also indicate that the residuals of the model are typically dispersed, as shown in Figure 2. Figure 3 shows that the residual plot designates homoscedasticity. The residual is randomly distributed without systematic shapes.

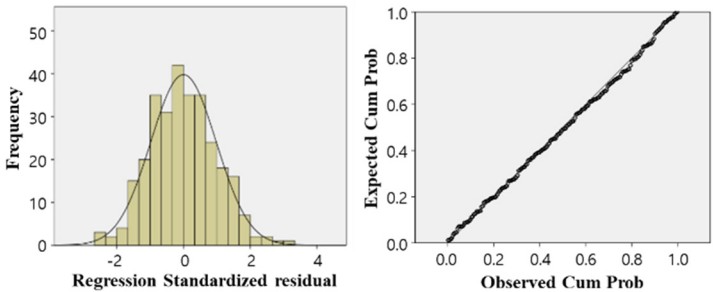

**Figure 2.** Histogram of residuals and Q–Q plot.

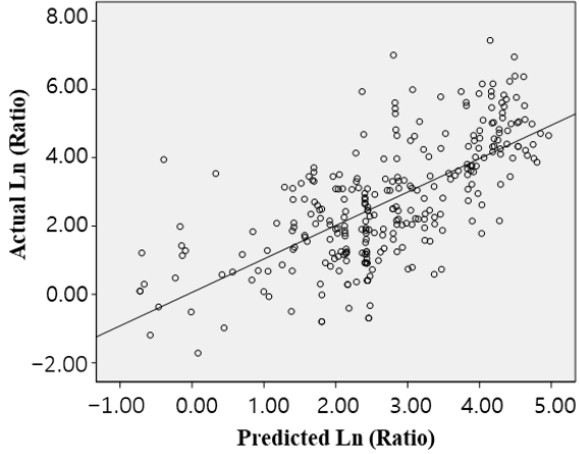

**Figure 3.** Residuals plot for the model.

This distribution reveals that the variances of the residuals are constant.

Finally, as shown in Figure 4, actual and predicted values are investigated. The scatter plot reflects the actual log-transformed loss ratio versus the predicted log loss ratio. The adjusted R-squared value of 0.471 indicates that 47.1% of the variability of the dependent variable can be described by the five independent indicators.

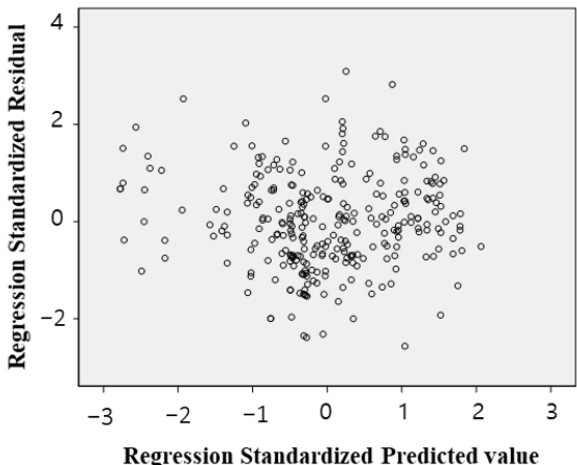

**Figure 4.** Actual vs. predicted value.

## 5. Discussion

For the construction management of a sustainable construction site, it is essential to identify and quantify losses according to scientific analysis. However, research on third-party loss was relatively insufficient, compared to that on material loss. To fill the gap, this study explored the third-party losses in actual construction sites of South Korea to define the variation with the material losses and to identify the interrelationship among losses and indicators for sustainable construction management. The severity and frequency of the significant loss causes were examined for each of these classifications to delineate the causes of loss.

The ANOVA analysis verifies that the mean loss ratios of third-party and material in the construction site are significantly different. The frequency of third-party losses is about 2.6 times higher than the material losses. The severity of third-party losses is about 4.5 times higher than the material losses. The third-party losses in construction sites are more severe than the materials losses in both frequency and severity. Therefore, in terms of construction site risk management, it is necessary to focus on the prevention of third-party loss.

As a final point, the loss indicators are defined based on the loss records. The critical indicators are an essential source for developing loss assessment models for building construction in South Korea. The multiple regression model is statistically significant, which confirms that the independent variables are able to predict the loss ratio.

The R-square, which indicates how well the data points match the approximation function, is the most commonly used estimator. The value of the adjusted R-square is 0.471, which means that the natural log-transformed dependent variable can be estimated by the significant variables with 47.1% of variability. Nonetheless, we hardly covered the remaining 52.9% of variability that we explained as undefined indicators in this study. Therefore, in order to have an R-square value of 50% or more (the normally acceptable minimum), further research is required. Further research is needed to improve the adjusted R-square, for example, by further discovering various indicators used in similar studies, such as project size, project type, and regional and seasonal characteristics. Additionally, more analysis is required by obtaining additional data, either from other insurance companies or from the public sector.

The *p*-values prove that five variables are statistically significant: (1) structural type, (2) total months, (3) aboveground floors, (4) underground floors, and (5) tropical cyclones. The structural type of building and the loss ratio have a positive relationship. The reinforced concrete work causes less third-party loss than other construction work, which is contrary to past findings that investigated the risk within the construction site [25]. The reason for this is that the main processes of reinforced concrete work, such as formwork assembly, reinforcing steel assembly, and concrete construction, are mainly carried out for the purpose of construction. Therefore, reinforced concrete construction is less damaging to third-parties around the construction site.

However, this study strengthens previous research that found that there is a significant relationship between the structural type of construction and the risks of the construction site. The total length of the construction period and the loss ratio have a negative connection, which means that as the total length of the construction period increases, the loss ratio decreases. This result improves the earlier study that found that the total length of construction was a significant factor for quantifying the amount of loss at construction sites [24]. The number of aboveground floors of the building and the loss ratio are negatively linked. As the aboveground floor number of the building increases, the loss ratio decreases. This study fortifies the prior research that there is an essential relationship between the number of aboveground floors in the building and the losses at construction sites [20]. The number of underground floors and the loss ratio have a negative linking, which proves that as the number of underground floors increases, the loss ratio decreases. This study supports the preliminary research that the number of underground floors is a critical factor for measuring the amount of loss at construction sites [20]. The risk grade of tropical cyclone and the loss ratio have a positive relationship. As the risk grade of tropical cyclone increases, the loss ratio increases. This strengthens past research that found that the risk grade is significantly related to the risk of construction and can be adopted to determine the risk [1,26].

## 6. Conclusions

The necessity of third-party loss assessment models for sustainable construction projects is increasing in concert with growing losses that are increasing in frequency due to the growth in scale and complexity of current construction projects and rapid urbanization. A numerical analysis of third-party loss is required to identify the loss indicators of construction sites to develop financial loss assessment models. Therefore, to meet the demand, this study adopts the ANOVA test and multiple linear regression analysis to explore the third-party loss in building construction sites utilizing claims payouts, as a case study in South Korea.

Through statistical analysis, even though the R-square value is relatively low, five indicators influencing third-party loss were identified. With the five indicators and the model, project holders, insurance companies, and construction companies can estimate monetary losses for sustainable construction projects and reduce losses through management of the five indicators. In addition, other countries whose construction environment and natural disasters are somewhat similar to those of Korea will be able to estimate losses by applying the results and structure of this study.

However, this study examined claim payouts incurred by one insurance company in South Korea. Although the model can be applied at construction sites similar to South Korea, if the environment at the construction site is dissimilar, the model will have to be further investigated. Moreover, the authors were unable to control regional or seasonal characteristics in the analysis due to data availability. Results may be biased depending on regional or seasonal characteristics; therefore, further research is needed to obtain relevant information from open sources or obtain additional data and merge it into the regression data set. Hence, we need further research through the acquisition of loss data from other countries and the data from different insurance companies or public sectors.

**Author Contributions:** Conceptualization, J.-M.K.; Data curation, K.-C.H., S.A.; Funding acquisition, J.-M.K.; Investigation, K.S., J.-M.K.; Methodology, J.-M.K., S.S.; Software, S.A., S.S.; Validation, S.S., K.-C.H.; Writing—original draft, K.S.; Writing—review and editing, J.-M.K., K.S. All authors have read and agreed to the published version of the manuscript.

**Funding:** The Research Funds of Mokpo National University in 2019 supported this research.

**Conflicts of Interest:** The authors declare no conflict of interest.

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
