# Peer review of "Quantifying the Third-Party Loss in Building Construction Sites Utilizing Claims Payouts: A Case Study in South Korea"

_sustainability, doi:10.3390/su122310153_

Round 1

Reviewer 1 Report

Dear Authors,

unfortunately, the paper has a number of shortcomings due to which it is not possible to accept it for publication in this form.
The extent of the necessary changes cannot be subsumed under major revision. For this reason, significant substantive changes are necessary for the paper to be considered for possible publication. The key shortcomings and explanations are listed below:

1. Research framework

The main shortcoming is a lack of clear research framework that should contain a clearly set main research goal and objectives, research methodology and research hypothesis. 

2. Data collection

The database of collected data is not clearly described.
The number of collected data (cases) and their structure and possible conditions and constraints in collecting are missing. They are not clearly described. 

3. Data Analysis

Before describing statistical indicators, an introductory description of the indicators themselves as well as the purposefulness of the presentation of the indicators in the context of the goal of the overall research would be useful.

The presentation of indicators in the tables 1 and 2 is deficient. The descriptions under Category (Material - ?, Third-party - ?, Min (?) , Max (?), Average (?)) are incomplete and unclear. 

It is somehow usual to state which software has been applied.

Here we come to another problem. It is stated: "...a statistical validation is required to analyze the significant difference between the risk arising from losses, i.e., third -party and material, of construction sites as a systematic approaching." Namely, it has been noticed before, and here it is now confirmed that nowhere is the connection between risk and losses clearly established and described. Do losses arise as a result of risks or do risks arise from losses or is both possible? Which here means "..as a systematic approaching"? In addition, it is not explained why the term third-party losses is used. It is probably taken because it is used in insurance companies, but in the context of research this requires an explanation, and perhaps a name change.

4. Investigation of third-party losses

It is stated: "We grouped the third-party losses by seven loss causes, such as failure of construction, typhoons, heavy rain,..., etc."  (???) The authors stated that it is about seven loss causes, after it list six of them, and then stands etc.

After Descriptive statistics table the Authors state that Figure 1 represents an analysis of the causes of loss. But what kind of analysis is this? How was it implemented? Nowhere has this been explained before? Moreover, it is not clear by what criteria and in what way the formed zones are represented in the coordinate system. They may be such, but the criteria for selecting values ​​should have been explained, at least in essence.

5. Multiple regression analysis

The authors begin with: "This study employs a multiple-regression analysis approach to define the interrelationship among the loss and indicators to create a loss estimation model according to the losses."

It stands below:"This research defines two categories, construction information and natural disasters as independent variables..."

There is no explanation as to how exactly these independent variables were selected. Seven types of third-party losses have been considered previously.

There is no word on the previous analysis that would select the just mentioned indicators.

The name of the subchapter is Loss indicators. Risk indicators are described in the description and in the table.

6. Results

The authors did not clearly define the accuracy criteria for accepting the proposed model. When it comes to R2 as the most commonly used estimator which indicates how well the data points match the approximation function, i.e. as a measure of the general match of the model, here it is very low, below the normally acceptable minimum of 50 %.

7. Conclusions

The Authors point out: "These findings and results provide a vital reference to insurance companies, ..."

But, as the most important it is necessary to point out that it is not clear what the result of the research is. Essentially, we do not know how the five indicators were selected, and using them in the analysis we obtain an unacceptably low model accuracy.

8. English language

Extensive editing of the English language is indispensable.

Reviewer 2 Report

This paper aims at quantifying the losses to third-party in the construction sites by  determining the loss indicators and identifying the relationship between the losses and the indicators  for improving the sustainability on the building construction sites.
I thing, that the scope and purpose of the article may be of interest to the reader.
The way of the presented analyzes suggests that this is the initial phase of research. This article makes an interesting contribution to the topic of construction losses and risks.
An ANOVA analysis and  multiple regression analysis are adopted to identify the variance. 
The purpose of regression is to identify the key factors influencing the main variable (loss) and to determine their degree of impact (by loss Indicators).
 These factors should be as little related to each other as possible and strongly correlated with the loss variable.
I can see the lack of analysis of the relationships between variables. This is crucial when selecting variables for the regression model. The authors took into account many variables. Variables should be used in stages. When adding another factor, check the R-square level.
The scope of application of both methods has to be expanded and presented in more clear for better understanding of the problem by the reader.
How to use ANOVA to analyze data is presented in the paper for example: Dziadosz, A., & Rejment, M. (2019, July). Risk assessment in construction project using statistic approach. In AIP Conference Proceedings (Vol. 2116, No. 1, p. 180012). AIP Publishing LLC. There are quite a few articles on how to use regression and interpret the results in detail. I recommend extending your references. The R-square value is not quite high (only 0.471). The authors rightly emphasize that 52,9% of the losses on the construction site could not be explained. I propose to indicate further research directions on such an interestingly presented problem. I also suggest changing the title of the article, it is not adequate to the content.
The comments do not minimize from the value of the work/paper and the authors' contribution to it.

Author Response

Please see the attched file.

Reviewer 3 Report

  1. The authors mention that the third-party losses happen outside the field from construction activities, but still need to define the term in a formal way. Can the authors cite any formal definition of the third-party loss from literature? What are the detailed examples of the loss? Table 3 shows seven “causes” of the third-party losses but still needs to provide what kinds of losses are associated with those causes.
  2. ANOVA: the paper does not show the number of projects (observations) in each group (third-party vs. material). From the regression analysis and Table 6, I assume there are 310 projects in the third-party category. Do the authors have a comparable number of observations for the material group? Can the authors show the numbers in Table 3?
  3. The current forms of the ANOVA and regression analysis fail to control project types, regional and seasonal characteristics. The models should control those variations to avoid biased estimates. For instance, it can be done by adding fixed effects in the model. Furthermore, the paper is missing the regression equation.

Author Response

Please see the attched file.

Reviewer 4 Report

The reviewer appreciates the effort made by the authors to try to provide a tool designed by the scientific community that will undoubtedly be an invaluable aid in minimizing the risks of third parties and allow greater control throughout the construction process. The article has a correct structure, although the presentation and the detail of the content in an important part of it can be improved. The reviewer understands that, being a field of work where most of the data come from insurance companies, it is not easy to have them as if they were an academic database; but, the authors must know how to combine this availability of data with an adequate presentation to the scientific community. That is why the reviewer considers that, for the work to achieve the desired impact, the authors must face a series of revisions that are dictated below:

1) INTRODUCTION. The introduction must be rethought, providing it with a better narrative that, based on all the studies available in the field of risk in construction works, clearly details the interest and need for this study with a clear approach to the objectives of the work.

2) DATA COLLECTION. In this part of the article, the data, the type and the origin of the data should be detailed in order to understand the use of these data in the authors' subsequent study. It is not clear what kind of data is being used next.

3) CONCLUSIONS. The conclusions that the authors draw from the results of the work are more of a declaration of intent; when they should be a detailed exposition, the result of comparing, among other things, the final work with the objectives set and the contributions that the study provides to the scientific community and the professional sector.

4) REFERENCES. It is recommended to review the way in which the citations are included in the article, since it is not easy to find them; if they are related to the citation number in the REFERENCES section it will be much easier to compare them. The references provided are clearly scarce. The reviewer considers that, taking into account the proposed revisions, the number of references in the work will be greater, giving greater strength to the final result.

Author Response

Please see the attched file.

Round 2

Reviewer 1 Report

Research framework still does not reach the required scientific rigor standards. The main research goal should be still more precisely defined. What are the objectives?

Section 1.2 should contain a clearly set main research goal (not goals, one is the main research goal) and objectives. Please, list them precisely.

1.3  Research methodology section is not written in an acceptable and usual way in terms of scientific paper requirements (should be precise, methods that are applied and methodological steps, not descriptions with examples). Please, be precise regarding applied software.

Research hypothesis has not been established, again. It has to be established precisely.

2.1 Data availability

What are martial losses? Please, explain „insurance conditions“.

3.1 Descriptive statistics and ANOVA analysis

Please add the units in the table, as well as the version (year of software).

Please, provide the full title of Table 2. Redesign the content of the pasus ""Table 2.    were significantly different" in a more precise way. (pg. 5/13)

Is it possible to enlarge Figure 1. for better readibility and visibility?

4. Multiple regression analysis

Please, be more precise and clear in definition of predictors and dependent variable, distinction between risk and loss in the terms of the whole paper context. Maybe I'm wrong but it seems that the term indicator  is used the first time in the section 4. It should be explained.

4.2 Results

R2 is again below the acceptable value. 

Conclusions should be still adjusted taking into account that very important fact in the terms of more precisely established guidelines for further research.

Reviewer 3 Report

I appreciate the authors revising and resubmitting the paper based on the reviewers' comments.

The authors mention that regional or seasonal characteristics cannot be controlled in the analysis due to data availability. However, the authors should be able to obtain the missing information from public sources and merge it into their regression dataset. Please reconsider controlling for the regional factors. I believe this is important to avoid biased estimation. 

The newly-added regression equation should be fixed to be a formal equation, meaning that it is missing an error term. 
